# 5-ALA Improves the Low Temperature Tolerance of Common Bean Seedlings through a Combination of Hormone Transduction Pathways and Chlorophyll Metabolism

**DOI:** 10.3390/ijms241713189

**Published:** 2023-08-25

**Authors:** Xinru Xue, Minghui Xie, Li Zhu, Dong Wang, Zeping Xu, Le Liang, Jianwei Zhang, Linyu Xu, Peihan Zhou, Jianzhao Ran, Guofeng Yu, Yunsong Lai, Bo Sun, Yi Tang, Huanxiu Li

**Affiliations:** College of Horticulture, Sichuan Agricultural University, Chengdu 611130, China; xuexinru1998@163.com (X.X.); 13603489353@163.com (M.X.); 18784009622@163.com (L.Z.); wd814702541@163.com (D.W.); 17882009408@163.com (Z.X.); 15294410767@163.com (L.L.); zhangjw1831@163.com (J.Z.); 15517900278@163.com (L.X.); zph2022718@163.com (P.Z.); rjz123456@139.com (J.R.); 18330735779@139.com (G.Y.); laiys@sicau.edu.cn (Y.L.); bsun@sicau.edu.cn (B.S.); tangyi@sicau.edu.cn (Y.T.)

**Keywords:** 5-aminolevulinic acid, cold stress, antioxidation system, hormone signal transduction, porphyrin anabolism, chlorophyll anabolism

## Abstract

Low-temperature stress is a key factor limiting the yield and quality of the common bean. 5-aminolevulinic acid (5-ALA), an antioxidant in plants, has been shown to modulate plant cold stress responses. However, the molecular mechanisms of 5-ALA-induced physiological and chemical changes in common bean seedlings under cold stress remains unknown. This study explored the physiological and transcriptome changes of common bean seedlings in response to cold stress after 5-ALA pretreatment. Physiological results showed that exogenous 5-ALA promotes the growth of common bean plants under cold stress, increases the activity of antioxidant enzymes (superoxide dismutase: 23.8%; peroxidase: 10.71%; catalase: 9.09%) and proline content (24.24%), decreases the relative conductivity (23.83%), malondialdehyde (33.65%), and active oxygen content, and alleviates the damage caused by cold to common bean seedlings. Transcriptome analysis revealed that 214 differentially expressed genes (DEGs) participate in response to cold stress. The DEGs are mainly concentrated in indole alkaloid biosynthesis, carotenoid biosynthesis, porphyrin, and chlorophyll metabolism. It is evident that exogenous 5-ALA alters the expression of genes associated with porphyrin and chlorophyll metabolism, as well as the plant hormone signal transduction pathway, which helps to maintain the energy supply and metabolic homeostasis under low-temperature stress. The results reveal the effect that applying exogenous 5-ALA has on the cold tolerance of the common bean and the molecular mechanism of its response to cold tolerance, which provides a theoretical basis for exploring and improving plant tolerance to low temperatures.

## 1. Introduction

The common bean (*Phaseolus vulgaris* L.) is one of the most important vegetable crops for human nutrition globally, providing protein, vitamins, and minerals for the human diet [1]. The early spring production of common beans is often limited by low temperatures, which cause limited root growth, leaf wilting, and even death. This leads to an overall decline in common bean yield and even mass crop failures, causing serious economic losses. Therefore, studies need to explore the physiological and molecular mechanisms of common beans under low-temperature stress, with the overarching goal of improving their low-temperature tolerance.

Cold stress causes huge losses to global agriculture every year and seriously threatens food security. Cold stress leads to the accumulation of reactive oxygen species (ROS) and malondialdehyde (MDA) in plants, thereby breaking the original dynamic balance, which results in the oxidation of biofilm, protein, and nucleic acids, and damages plant tissues or causes the death of plant cells [2,3]. Superoxide dismutase (SOD), peroxidase (POD), and catalase (CAT) antioxidant enzymes are important components of plant metabolism [4]. Studies have demonstrated that enzymes regulate plant metabolism—such as plant hormone signal transduction, glutathione metabolism [5], and β-alanine metabolism [6]—to remove excess ROS. Reference [7] found that antioxidant enzymes also play a role in plant responses to abiotic stresses induced by pesticides. Proline and other low-molecular-weight solutes act as osmoregulation substances to protect cell osmotic pressure, thereby reducing the damage caused by low-temperature stress in plants [4].

As the necessary biosynthetic precursor of all heterocyclic tetrapyrrole molecules, 5-aminolevulinic acid (5-ALA) is a potential plant growth regulator [8]. It has been reported that exogenous 5-ALA can regulate plant growth and development, nutritional growth, seed germination, and fruit coloring [9]. In addition, 5-ALA regulates some metabolic processes, such as the biosynthesis of chlorophyll, heme, and siroheme [10], and alleviates abiotic stress by regulating photosynthesis, nutrient absorption, antioxidant defense, and osmotic regulation. Priming with 5-ALA can enhance plant tolerance to abiotic stresses including salinity [11], drought [12], heat [13], cold [14], and UV-B [15].

To date, it is unclear whether exogenous 5-ALA can improve cold tolerance and promote the growth of common bean seedlings under cold stress by affecting some physiological and biochemical processes. Herein, we explored changes in the biomass, antioxidant system, osmotic solute, and transcriptome of common bean seedlings after an exogenous application of 5-ALA under normal and cold stress conditions. We comprehensively analyzed the effects of exogenous 5-ALA treatment on the physiology and biochemistry of common bean seedlings under cold stress. Our findings provide new insights into the potential mechanism of 5-ALA-mediated low-temperature responses of common bean seedlings.

## 2. Results

### 2.1. Exogenous Application of 5-ALA Improved the Growth under Cold Stress

There were four treatment groups: (1) NT: seedlings treated with distilled water and grown in normal conditions; (2) NTA: seedlings pretreated with 5-ALA and then grown in normal conditions; (3) LT: seedlings treated with distilled water and then grown under cold stress; and (4) LTA: seedlings pretreated with 5-ALA and then grown under cold stress. There were differences in the plant phenotypes among the four treatments after 72 h of the experiment (Figure 1A). The normal temperature control plants grew well (NT and NTA), and 5-ALA promoted the growth of bean seedlings at normal temperature. LT plants were significantly wilted, had soft petioles, and the leaves curled and withered. LTA plants showed less severe symptoms under cold stress since only the leaves were slightly dehydrated, and the leaf edges were wilted. Biomass measurement showed that cold stress significantly reduced the growth of bean plants, whereas 5-ALA pretreatment alleviated this effect (Figure 1B–D). The plant height, stem diameter, fresh aboveground weight, and dry and fresh underground weight of LT plants were significantly lower than those of NT plants at 72 h of cold stress. However, the aboveground dry and fresh weight and underground fresh weight of LTA plants increased by 36.76%, 38.88%, and 38.46%, respectively, compared to the LT plants. These results suggest that 5-ALA pretreatment could improve the tolerance of common bean seedlings to cold stress. 

### 2.2. Exogenous 5-ALA Application Alleviated Increases in Relative Conductivity and Proline under Cold Stress

The proline content in the leaves gradually increased with the extension of low-temperature stress time. LT and LTA plants reached significant levels 24 h after stress. After 72 h of cold stress, the proline content of LT plants was 2.21 times that of NT plants, whereas the proline content of LTA plants was 1.32 times that of LT plants (Figure 2A). With the extension of cold stress time, the relative conductivity (REC) of LT plants first showed a decreasing trend, and then increased, but it was always significantly higher than that of NT plants. However, the REC of LTA plants was significantly higher than that of NT plants at 72 h. After 72 h of cold stress, the REC of LT plants was 1.69 times that of NT plants, and the REC of LTA plants was significantly reduced by 23.83% compared to the LT plants (Figure 2B).

### 2.3. Exogenous 5-ALA Application Enhanced Antioxidant Enzyme Activity and ROS Scavenging under Cold Stress

A ROS staining experiment was performed to explore whether 5-ALA affects ROS status. The results of DAB (Figure 3A) and NBT (Figure 3B) staining confirmed that ROS increased significantly in LT plant leaves. With the extension of stress time, dark brown and blue polymer products increased gradually, indicating that H_2_O_2_ and O^2−^ gradually accumulated in the leaves under cold treatment. On the other hand, the color change of LTA plant leaves was not obvious, which indicated that the 5-ALA treatment significantly reduced the accumulation of H_2_O_2_ and O^2−^ in leaves under cold stress.

Compared to control plants, the cold treatment significantly increased the activities of SOD, POD, and CAT enzymes in leaves. With the extension of stress time, POD and CAT first showed an increasing trend, and then decreased, whereas SOD showed a trend of first decreasing and then increasing. After 72 h of stress, the activities of SOD, POD, and CAT in LT plant leaves were 1.87 times, 1.41 times, and 1.23 times that of NT plants, respectively. The activities of SOD, POD and CAT in the leaves of LTA plants were higher than those of LT plants at all time points. Notably, the activity of SOD increased rapidly in the late stage of cold stress, whereas the activity of CAT increased rapidly in the early stage of cold stress. At the 72 h time point, the activities of SOD, POD, and CAT in the leaves of LTA plants were 1.3 times, 1.12 times, and 1.1 times that of NT plants, respectively (Figure 3C–E). Under cold stress, the content of MDA in the leaves of common bean seedlings increased sharply. After 72 h of stress, the content of MDA in LT plants was 2.29 times that of NT plants. Moreover, the content of MDA in LTA plants was significantly lower than in LT plants, and did not fluctuate during the whole treatment period (Figure 3F).

### 2.4. Quantitative Assessment of Transcriptome Data and qRT-PCR Validation

Twelve cDNA libraries were sequenced from the four treatments. After quality control, 75.75 Gb of clean data was obtained, and the percentage of Q30 base of each product was not less than 93.42%. These reads were then mapped to the common bean reference genome.

To confirm the RNA-seq results, 12 genes were randomly selected for qPCR. The relative gene expression obtained from qRT-PCR was calculated using the 2^−∆∆Ct^ method. As shown in Figure 4, the comparisons showed that the 12 genes examined by qRT-PCR had similar expression profiles as in the RNA-seq analysis, indicating that reliable RNA-Seq data were obtained from the samples.

### 2.5. Identification of Differentially Expressed Genes (DEGs) and Analysis of Expression Pattern

The three different comparisons found that NT and NTA had 89 upregulated genes and two downregulated genes, NT and LT had 3624 upregulated and 2753 downregulated genes, and LT and LTA had 106 upregulated and 108 downregulated genes (Figure 5A). In addition, 37 upregulated and two downregulated genes were specifically expressed in NT compared to NTA, 3555 upregulated and 2724 downregulated genes were specifically expressed in NT compared to LT, and 61 upregulated and 79 downregulated genes were specifically expressed in LT compared to LTA. In these three comparisons, two genes were generally regulated (Figure 5B).

Based on K-means clustering analysis, the various genes expressed in the leaves of seedlings administered with the above treatments were clustered into 14 groups (Figure 5C). The analysis showed the gene expression changes of leaf samples under the different treatments, which were further subdivided into six expression modes. Specifically, 1437 genes were upregulated from NT to LT and downregulated from LT to LTA; they were further divided into three subgroups (subgroups 4, 5, and 10) according to their trend amplitude. A total of 243 genes were downregulated from NT to LT and upregulated from LT to LTA (subgroup 12); 1122 genes were continuously upregulated from NT to LTA (subgroups 7 and 14); 1484 genes were continuously downregulated from NT to LTA (subgroups 8, 11, and 13); 1080 genes were downregulated from NT to LT, and the expression of LT to LTA did not change significantly (subgroup 9). Furthermore, 1145 genes were upregulated from NT to LT, whereas the expression of LT to LTA did not change significantly (subgroups 1, 2, 3, and 6).

### 2.6. GO and KEGG Analysis of DEGs among Different Treatments

A gene ontology (GO) analysis was conducted to obtain functional annotations of DEGs for each comparison. The DEGs in both the NT vs. LT and the LT vs. LTA comparisons were annotated and classified according to the biological process (BP), molecular function (MF), and cellular component (CC) terms in line with the GO database The primary categories in BP were metabolic, cellular, and single-organism processes; the prominent MF categories were binding, catalytic, and transporter activity; and the most abundant categories in CC were membrane, membrane part, and cell (Figure 6A,B).

We further mapped the DEGs to the Kyoto Encyclopedia of Genes and Genomes (KEGG) database to analyze the metabolic pathways. Between the NT and LT libraries, 2149 DEGs were assigned to 130 KEGG pathways, including taurine and hypotaurine metabolism, synthesis and degradation of ketone bodies, limonene and pinene degradation, circadian rhythm—plant, and isoflavonoid biosynthesis, which were the top five significantly enriched pathways. Between LT and LTA libraries, 88 DEGs were assigned to 58 KEGG pathways, including indole alkaloid biosynthesis, glucosinolate biosynthesis, carotenoid biosynthesis, porphyrin and chlorophyll metabolism, and propanoate metabolism, which were the top five significantly enriched pathways (Figure 6C,D).

### 2.7. Photosynthesis and Porphyrin and Chlorophyll Metabolism Response to Cold Stress under 5-ALA Applications

The differential genes related to photosynthesis and chlorophyll synthesis in the transcriptome analysis of exogenous 5-ALA-mediated responses of common bean seedlings to cold stress were analyzed, and 17 DEGs were identified. In the carbon fixation pathway of photosynthesis, the exogenous application of 5-ALA under cold stress induced downregulation of malate dehydrogenase expression and upregulated expression of two genes encoding phosphoenolpyruvate carboxylase (Figure 7a). In photosynthesis, it was found that the expression of *PsbB* and *Psb28* genes in photosystem II was upregulated, the expression of the PetA protein in the cytochrome b6/f complex was downregulated, the expression of subunit PsaA in photosystem I was downregulated, and the expression of subunit PsaN was upregulated (Figure 7b). In porphyrin and chlorophyll metabolism, we observed that the expression of differential genes encoding magnesium chelatase and magnesium-free chloroplast oxygenase was upregulated, and the expression of differential genes encoding protochlorophyll reductase, erythrophyllin catabolic reductase, chlorophyllase, and chlorophyll reductase was downregulated. The upregulated gene is related to the red chlorophyll metabolite, which is regulated by a negative feedback loop to promote the accumulation of chlorophyll a and b (Figure 7c). Divinylchlorophyll a and chlorophyll a are the main components of chlorophyll synthesis. Altogether, these results suggest that the reduction in the chlorophyll level was inhibited by maintaining the level of the main substrates of chlorophyll biosynthesis (divinylchlorophyll a and chlorophyll a).

### 2.8. Plant Hormone Pathway Response to Cold Stress under 5-ALA Applications

DEGs associated with hormone signal transduction in response to cold stress were determined after transcriptome analysis of common bean seedlings after exogenous 5-ALA, mainly in ethylene, abscisic acid, gibberellin biosynthesis and brassinosterol, and jasmonic acid signal transduction (Figure 8).

Figure 8 shows that abscisic acid biosynthesis can be divided roughly into three steps: from zeaxanthin to xanthoxin in chloroplasts, which is then converted to ABA through ABA aldehyde using a catalytic reaction. In this process, six DEGs were identified in leaves, including one expression upregulated gene and one expression downregulated gene encoding zeaxanthin epoxidase, as well as two upregulated and two downregulated genes encoding xanthotoxin dehydrogenase. Ethylene biosynthesis is mainly divided into three steps. First, methionine is transformed into S-adenosylmethionine (SAM) with the participation of adenosine triphosphate (ATP), and is then split into 1-aminocyclopropane 1-carboxylic acid (ACC) and methylthioadenosine (MTA). Finally, ACC is transformed into ethylene (ETH) with the catalysis of the ethylene-forming enzyme (EFE). During this process, two differentially expressed genes were identified to be downregulated in the leaves, including one encoding methionine adenosyltransferase and one encoding aminocyclopropane carboxylate oxidase.

Furthermore, Figure 8 shows that in the abscisic acid signal transduction pathway, one DEG encoding *PP2C* was upregulated. A differentially expressed gene encoding *ERF1/2* was downregulated in the ethylene signal transduction pathway. In the brassinosterol signal transduction pathway, it was found that eight differentially expressed genes encoding the *BAK1* receptor were upregulated, four genes encoding the *BAK1* receptor were downregulated, and four genes encoding *BRI1* were upregulated, which fully activated the downstream BR signal, further relieving the inhibition of *BZR1/2* and activating the transcription regulation of the downstream by the *BAR1/2* family. In the jasmonic acid signal transduction pathway, one DEG encoding *COI1* was upregulated, one DEG encoding *JAZ* was upregulated, two DEGs encoding *MYC2* transcription factors were downregulated, and one DEG encoding *MYC2* was upregulated.

## 3. Discussion

Low temperature, a common abiotic stress, severely limits crop growth and yield [16]. 5-ALA, an important precursor of the tetrapyrrole biosynthesis pathway, is a plant growth regulator that regulates plant defense mechanisms to mitigate the harmful effects of abiotic stress [9,17]. This study explored the physiological and molecular mechanisms through which 5-ALA pretreatment alleviates cold stress using the common bean as the study material. It was found that 5-ALA pretreatment promoted the adaptation of common bean to cold stress by regulating physiological responses. Transcriptome analysis was applied to identify a series of pathways that respond to 5-ALA applications, such as hormone signal transduction, photosynthesis, and chlorophyll synthesis, to improve the tolerance of common bean (Figure 9). Specifically, we focused on comparing LT and LTA to determine the effects of 5-ALA on cold tolerance.

Cold stress leads to cell dehydration [18]. In most cases, cold stress results in the closure of plant stomata and a reduced photosynthesis rate, leading to phenotypes such as wilting and leaf curling [19]. We found that 5-ALA pretreatment alleviated the decline of water content under cold stress, and the phenotype of the common bean was closer to its normal growth state, consistent with the results observed in pepper plants [20]. In addition, it was found that 5-ALA pretreatment alleviated the growth inhibition induced by cold stress. 5-ALA-induced growth improvement has been observed in many plants under stress, such as Chinese cabbage [21], rape [22], tomato [14], and cucumber [23]. These studies attributed the growth improvement to the chlorophyll synthesis triggered by 5-ALA, the absorption of essential nutrients, including N, P, and K, the accumulation of osmotic substances (glycine, betaine, and proline), the activity of antioxidant enzymes (SOD, POD, and CAT), and the increased photosynthetic rate [24,25,26].

Cold stress causes serious membrane lipid peroxidation, increases membrane permeability, causes cell metabolism disorders, and increases the content of permeable substances [27,28]. Conductivity can indirectly evaluate cell membrane disorders by quantifying the electrolytes released into the water after tissue infiltration [29]. Free proline, an osmotic protector and antioxidant, maintains a cell environment suitable for different metabolic processes, and enhances stress adaptability [30]. In this study, the REC and the accumulation of the proline level of bean plant leaves increased under cold stress. A similar result was found in tomatoes [31]. Dehydration stress is also associated with cold stress. The decline in metabolic activity, the interruption of leaf transpiration, and the inhibition of water absorption by roots and shoots are the main reasons for the dehydration stress of plants under cold stress. Alet et al. [32] reported that the putrescine accumulation of oat ADC transgenic plants was associated with higher proline accumulation and therefore improved tolerance to dehydration and freezing stress. This study revealed that 5-ALA treatment could induce more proline accumulation and alleviate the dehydration stress of common bean seedlings. It should be noted that the proline accumulation in the common bean induced by 5-ALA may not be the result of stress injury, but may be an active osmotic solute accumulation. It may be due to the increase of exogenous ALA content that the metabolism of glutamate is regulated in a feedback manner from the chlorophyll synthesis pathway to the proline pathway [33]. The accumulation of a large amount of ROS under stress conditions will cause membrane lipid peroxidation in plants. A small amount of ROS can be used as a signal substance to activate some metabolic pathways, thus enhancing the ability of plants to adapt to stress [34,35]. The content of MDA, a common product of lipid peroxidation, reveals the extent of oxidative damage to plants. To mitigate these harmful effects, plants have evolved a defense system that removes these toxic substances through the antioxidation of enzymatic and nonenzymatic systems [36,37]. In this study, ROS such as O^2−^ and H_2_O_2_ led to membrane lipid peroxidation, and the MDA content increased under cold stress. However, these effects were alleviated after 5-ALA pretreatment, suggesting that the activities of antioxidant enzymes (SOD, POD, and CAT) in common bean seedlings increased. It has previously been reported that 5-ALA alone cannot eliminate active oxygen [38], and O_2_ can be rapidly converted into H_2_O_2_ by SOD [39] and converted into H_2_O or O_2_ by either the AsA or GSH regeneration cycle or CAT [40]. Therefore, the antioxidant system may play a key role in ROS reduction. Previous studies reported that applying 5-ALA activated plant defense systems and defense-related genes—such as genes encoding SOD, POD, and CAT—in rice and strawberry plants under osmotic and photodynamic stress, and reduced the excessive production of ROS and MDA [41,42,43]. It is worth noting that 5-ALA is the precursor of heme biosynthesis. CAT, POD, and APX contain heme cofactors [11], which may explain why the antioxidant enzyme activity in 5-ALA-treated seedlings is stimulated. 5-ALA pretreatment may also start a nonenzymatic system, regulate the expression of glutathione (GSH) and ascorbic acid genes, and inhibit the accumulation of ROS [21]. In addition, many scholars believe that exogenous 5-ALA inhibits the accumulation of H_2_O_2_ in plant tissues under stress [22,44], which is consistent with our research. However, some studies found that exogenous 5-ALA promoted the accumulation of H_2_O_2_ because H_2_O_2_, as a cell signal, improved plant stress resistance [45,46]. This also suggests that the tissue-specific regulatory mechanism of 5-ALA on H_2_O_2_ deserves further study.

Plants have evolved strategies to cope with cold stress by coordinating cold and hormone-signaling pathways [47]. Previous studies have shown that 5-ALA triggers increased tolerance to cold stress by regulating the biosynthesis of classic plant hormones, such as brassinosteroids (BR), cytokinins (CKs), and abscisic acid (ABA) during cold stress [17]. Under abiotic stress, ABA accumulates in plants through stress signal factors and regulates many important physiological and biochemical processes to improve the adaptability of plants to stress [48]. This study found that genes encoding the ABA biosynthetic protein (ABA2 expression was downregulated and SDR expression was upregulated) and genes associated with signal transduction (PP2C expression was upregulated) were differentially expressed after exogenous 5-ALA pretreatment. In the overexpression strain of Arabidopsis thaliana, it was found that ABA2 (SDR1) catalyzes the conversion of xanthaldehyde to abscisic aldehyde [49]. At the same time, another study reported that the increase in SDR expression under moderate cold is conducive to the accumulation of *ginsenosides* [26]. Thus, it is evident that 5-ALA alleviates the cold stress of common bean by enhancing the accumulation of secondary metabolites and ABA-dependent signal pathways. In addition, ACO, the essential enzyme encoding the ETH biosynthetic protein, showed a lower expression level under cold stress, thereby inhibiting ethylene synthesis and the expression of ethylene response factor ERF1B. Consequently, the transcription factor of the ETH signal transduction pathway was downregulated. Moreover, 5-ALA pretreatment increased the expression level of BRI1/2 and BZR1/2, the key factors of the BR signaling pathway, and COI1 and JZA, the key factors of the JA signaling pathway, under cold stress. BZR1, the core transcription factor of the BR signal, acts on CBF1 and the upstream part of CBF2 to regulate their expression, thereby inducing the expression of the CBF regulator [50]. In Arabidopsis, it was found that BZR1 also regulates other COR genes uncoupled with CBF to regulate plant responses to cold stress [51].

Some studies have also reported that the application of exogenous 5-ALA increases the content of chlorophyll in plants under cold stress [17]. Mounting evidence suggests that chlorophyll content is an important physiological indicator of plant tolerance to cold stress [20,52]. A 5-ALA pretreatment could improve chlorophyll content by regulating genes involved in chlorophyll synthesis, the chlorophyll degradation pathway, and the chlorophyll cycle. We identified two genes (Phvul.007G252700.v2.1 Phvul.001G132200.v2.1) encoding magnesium chelatase (CHLH), which is a key site for regulating the branching of tetrapyrrole intermediates from common PIX intermediates through Chl [53]. Therefore, the upregulation of CHLH observed after 5-ALA pretreatment may increase the synthesis of Chl and ultimately increase the Chl content under cold stress. A previous study also found that 5-ALA increased the expression of CHLH in cucumber plants under salt stress [11]. CLH and NYC encode chlorophyll enzymes to participate in chlorophyll catabolism and catalyze stress-induced chlorophyll decomposition [54,55]. Considering that the expression levels of CLH and NYC decreased after 5-ALA pretreatment, we speculated that 5-ALA pretreatment may inhibit chlorophyll degradation. The non-heme oxygenase PTC52 gene, a kind of oxygenase with an inner plastid envelope, links the synthesis of Pchlide b with the introduction of pPORA [56]. Here, the expression of PTC52 increased after 5-ALA pretreatment, promoting chlorophyll synthesis under cold stress. Red chlorophyll decomposition metabolite reductase (RCCR) is a key enzyme that depends on the chlorophyll degradation pathway [57]. RCCR catalyzes the pheophorbide (pheide) into a primary fluorescent chlorophyll catabolite [58]. In the present study, the expression of RCCR decreased after 5-ALA pretreatment, which inhibited the catabolism of chlorophyll under cold stress.

The Psb28 protein, an important subunit of the PSII pigment-protein complex, is closely associated with water decomposition in light-dependent reactions. We found that the expression of CP47 increased under cold stress after exogenous 5-ALA treatment, and the increased expression of Psb28 augmented the electron transport rate and photochemical efficiency, thereby improving photosynthetic performance under cold stress. A similar result was observed in rice [59]. In this experiment, the expression of the gene encoding phosphoenolpyruvate carboxylase (PEPC) was upregulated after exogenous 5-ALA pretreatment. In C_3_ plants, malic acid formed through PEPC may participate in providing metabolic intermediates from the tricarboxylic acid cycle for stress survival [60], and CO_2_ fixation formed through PEPC may contribute to the Calvin cycle due to stomatal closure. However, further studies should determine whether this relates to the photosynthesis rate under cold conditions, as well as discuss the results and how they can be interpreted from the perspective of previous studies and of the working hypotheses. The findings and their implications should be discussed in the broadest context possible. Future research directions may also be highlighted.

## 4. Materials and Methods

### 4.1. Plant Materials

The common bean material, the cold-sensitive bean variety “Gluten-free green ground bean”, Seeds of “*Wujinlvdidou*”, a cold-sensitive common bean cultivar (Phaseolus vulgaris), was selected from the previous experiments of the Vegetable Research Laboratory, College of Horticulture, Sichuan Agricultural University, as the experimental material. Briefly, the seeds were soaked in tap water at 25 °C for 2 h and then placed at 25 °C to induce germination. After more than 50% of the seeds were germinated, they were sown in a nutrient bowl (9 cm × 9 cm) filled with a mixture of peat, perlite, and vermiculite (volume ratio 2:1:1), with one seed per nutrient bowl. The bowl was placed in an artificial climate chamber with the following environmental conditions: 12 h photoperiod, 25 ± 2 °C/18 ± 2 °C (day/night) temperature, 300 μmol·m^−2^·s^−1^ light intensity, and 70 ± 5% relative humidity.

### 4.2. Experimental Design

At the two leaves and one heart stage (10 days after germination), seedlings of uniform size were selected and divided into two groups for pretreatment. One group was sprayed with a 25 mg L^−1^ 5-ALA solution (25 mL per pot), while the other group was sprayed with the same volume of double-distilled water. After 24 h, the seedlings in each pretreatment group were further divided into two groups. The seedlings in one group were maintained under normal conditions, whereas seedlings in the other group were grown under cold stress at 4 °C. Notably, there were four treatment groups: (1) NT: seedlings treated with distilled water and grown in normal conditions; (2) NTA: seedlings pre-treated with 5-ALA and then grown in normal conditions; (3) LT: seedlings treated with distilled water and then grown under cold stress; and (4) LTA: seedlings pre-treated with 5-ALA and then grown under cold stress. The four treatments were subjected to a completely randomized block design.

Physiological indexes and transcriptomic experiments were performed with three biological replicates per treatment. Briefly, leaf samples were collected at 12, 24, 48, and 72 h after treatment for physiological experiments, and 24 h after treatment for transcriptome sequencing. All samples were immediately frozen in liquid nitrogen and stored in a refrigerator at −80 °C until subsequent analysis.

### 4.3. Determination of Biomass of Seedlings, Relative Conductivity and Proline

After 72 h of treatment, we measured the plant height and stem diameter of three plants per treatment. We then separated the aerial and root parts, and weighed them to obtain the fresh mass. The samples were dried at 105 °C for 15 min, followed by drying at 75 °C until reaching a constant weight to obtain the dry weight.

We soaked fresh complete leaves in a tube containing 20 mL of ddH_2_O to measure the initial electrical conductivity (L0). The test tubes containing the leaves in distilled water were kept at room temperature for 24 h, and L1 was determined. Next, we placed the test tubes in a boiling water bath for 20 min, allowed them to cool to room temperature, and then filtered and measured the boiled leachate for L2. We computed REC using the following formula [61]:REC (%) = [(L1 − L0)/(L2 − L0)] × 100%(1)

To determine the content of proline, fresh leaves (0.5 g) were homogenized in 5 mL of 3% sulfosalicylic acid for 10 min at 90 °C and then cooled for centrifugation. The supernatant (2 mL) was treated with 2 mL of glacial acetic acid and 3 mL of 2.5% acid ninhydrin, followed by 4 mL of toluene. The absorbance of the colored solution was measured on the UV-1800 spectrophotometer at 520 nm [62].

### 4.4. NBT and DAB Staining

To detect superoxide radicals, we used the nitrotetrazolium blue chloride (NBT) staining method, following the procedure described by Yang et al. [63]. We incubated leaves in NBT staining solution (0.5 mg/mL NBT, 10 mM potassium phosphate buffer, pH 7.6) in the dark for 6 h, then boiled them in 95% ethanol for 15–20 min before visualizing the blue precipitates. To detect hydrogen peroxide, we used the 3,30-diaminobenzidine (DAB) staining method. We incubated leaves in DAB staining solution (2 mg/mL DAB) in the dark for 6 h, then decolorized them as with NBT. The intensity of the brown coloration indicated the H_2_O_2_ content.

### 4.5. Quantification of MDA and Activities of SOD, POD, and CAT

The MDA measurement followed the procedure described by Chomkitichai et al. [64]. Leaf tissue (0.3 g) was ground with 5 mL of 10% trichloroacetic acid, and then centrifuged at 4000× *g* for 10 min. The resulting supernatant (2 mL) was mixed with 2 mL of 0.6% thiobarbituric acid (TBA). The mixture was boiled in water for 15 min and quickly cooled for centrifugation. The absorbance was measured at 532 nm, 600 nm, and 450 nm, and the MDA content was calculated.

To prepare the crude enzyme extract, 0.2 g of leaf powder was homogenized in 1.6 mL of cooled extraction buffer containing 0.05 mol L^−1^ phosphate buffer (SOD: pH = 7.8; POD, CAT: pH = 7.0). After homogenization, the leaf powder suspension was centrifuged at 12,000× *g* and 4 °C for 20 min, and the resulting supernatant was collected for subsequent tests.

The activity of SOD, POD, and CAT was determined according to the method of de Azevedo Neto et al. [65]. The activity of SOD (EC 1.15.1.1) was expressed as an enzyme activity unit of 50% inhibition of NBT photochemical reduction, and was measured at 560 nm; CAT (EC 1.11.1.6) activity was estimated by monitoring the reduction of absorbance at 240 nm during H_2_ decomposition; POD (EC 1.11.1.7) activity was calculated by monitoring the increase of absorbance at 470 nm when guaiacol is oxidized by H_2_O_2_ to release O^2−^, and the enzyme activity was calculated as a change of 0.01 per min of the absorbance value.

### 4.6. Library Construction, Sequencing and Data Analysis

The leaves of NT, NTA, LT and LTA 24 h after low temperature treatment were collected and RNA-seq was performed using the Illumina platform by Biomarker Technologies (Beijing, China). Three biological replicates were performed for each of the four sample groups. After cDNA library sequencing, clean data (clean reads) were obtained by removing reads containing adapter, reads containing ploy-N and low-quality reads from raw data. At the same time, Q20, Q30, GC-content, and sequence duplication levels of the clean data were calculated. The filtered reads were mapped to the common bean genome (https://phytozome-next.jgi.doe.gov/info/Pvulgaris_v2_1, accessed on 5 May 2022) sequence using HISAT2.

### 4.7. Identification and Analysis of Differentially Expressed Genes

FPKM (fragments per kilobase of coding sequence per million reads) was used to quantify the mapped gene expression levels. We performed differential expression analysis using DESeq2 1.6.3 software. The thresholds of FDR < 0.01 and |log2 (fold change)| ≥ 1 were used to identify differentially expressed genes (DEGs). Then, gene ontology (GO) enrichment analysis of the DEGs was performed using the Goseq R package [66]. KOBAS 3.0 software was used to test the statistical enrichment of DEGs in the Kyoto Encyclopedia of Genes and Genomes (KEGG) pathways [67].

### 4.8. Quantitative Real-Time PCR Validation

To verify the accuracy of the RNA-seq data, we selected 12 genes from the differentially expressed genes (DEGs) for qRT-PCR. Total RNA was extracted from the leaves of each treatment using the RNA prep Pure Plant Plus Kit (TaKaRa, Beijing, China) following the manufacturer’s instructions. The RNA concentration was determined using a spectrophotometer, and cDNA synthesis was performed using the PrimeScript™ RT reagent Kit with the gDNA Eraser kit (TaKaRa, Maebashi, Japan). The primers for these genes were designed using Primer3 (https://primer3.ut.ee/, accessed on 26 August 2022), and their specificity was confirmed using the NCBI database (https://www.ncbi.nlm.nih.gov/, accessed on 26 August 2022). qRT-PCR analysis was conducted using BIOMARKER (Biomarker 2× SYBR Green Fast qPCR Mix) on a CFX96 real-time PCR system (Bio-Rad, Hercules, CA, USA). We used Actin-11 as reference genes to normalize the data, and the relative expression data were calculated using the 2^−∆∆Ct^ method [68]. We performed three independent technical repeats with three biological replicates for the qRT-PCR analysis. The primers used in this study are listed in Appendix A.

### 4.9. Statistical Analysis

Statistical tests were performed with the SPSS software (version 26.0). Differences among treatments were determined using least significant difference (Duncan) tests, and the threshold for statistical significance was *p* < 0.05. Data were presented as the mean ± SD of three biological replicates.

## 5. Conclusions

This study revealed the mechanism through which exogenous 5-ALA enhances the cold resistance of the common bean after conducting physiological and comparative transcriptome analyses. Results indicated that 5-ALA pretreatment before cold stress increased the activity of proline and antioxidant enzymes, and decreased the content of REC, ROS, and MDA in seedlings. At the same time, it promoted growth and improved cold tolerance. This indicates that producers will be able to treat their bean seedlings with 5-ALA before the cold snap, thus protecting their crops from damage. However, the specific application method needs to be further explored. The transcriptome analysis identified the DEGs associated with the response to 5-ALA under cold stress. We found that exogenous 5-ALA enhanced cold tolerance by regulating the hormone signal transduction, chlorophyll metabolism, and photosynthesis of seedlings under cold stress. On this basis, we propose a new regulation model of 5-ALA to alleviate cold stress in common bean plants. Overall, our results have laid a foundation for further elucidating the mechanism of exogenous 5-ALA-mediated cold tolerance and provided a theoretical basis for improving the cold tolerance of common bean.

## Figures and Tables

**Figure 1 ijms-24-13189-f001:**
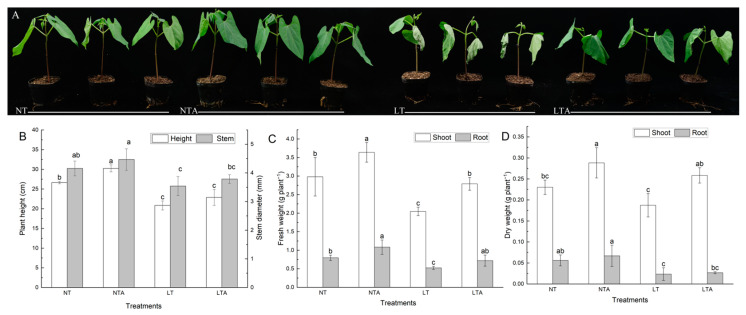
Effects of 5-ALA pretreatment on the phenotype. (**A**) plant height and stem diameter; (**B**) fresh weight; (**C**) dry weight; (**D**) plants exposed to cold stress. The plant phenotype is assessed 72 h after cold treatment. NT, distilled water plus normal conditions; NTA, 5-ALA plus normal conditions; LT, distilled water plus cold conditions; LTA, 5-ALA plus cold conditions. Each data point represents the mean of three replicate samples, and results are means ± standard deviation. Letters indicate significant differences at *p* < 0.05 by Duncan’s test.

**Figure 2 ijms-24-13189-f002:**
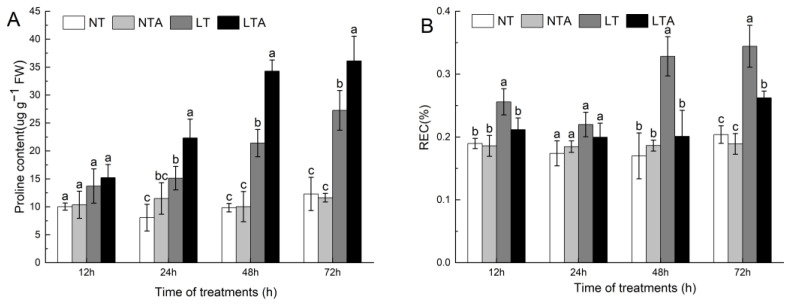
Effects of 5-ALA pretreatment on relative conductivity and proline under of plants exposed to cold. (**A**) Proline content (Pro); (**B**) Relative conductivity (REC). NT, distilled water plus normal conditions; NTA, 5-ALA plus normal conditions; LT, distilled water plus cold conditions; LTA, 5-ALA plus cold conditions. Each data point represents the mean of three replicate samples, and results are means ± standard deviation. Letters indicate significant differences at *p* < 0.05 by Duncan’s test.

**Figure 3 ijms-24-13189-f003:**
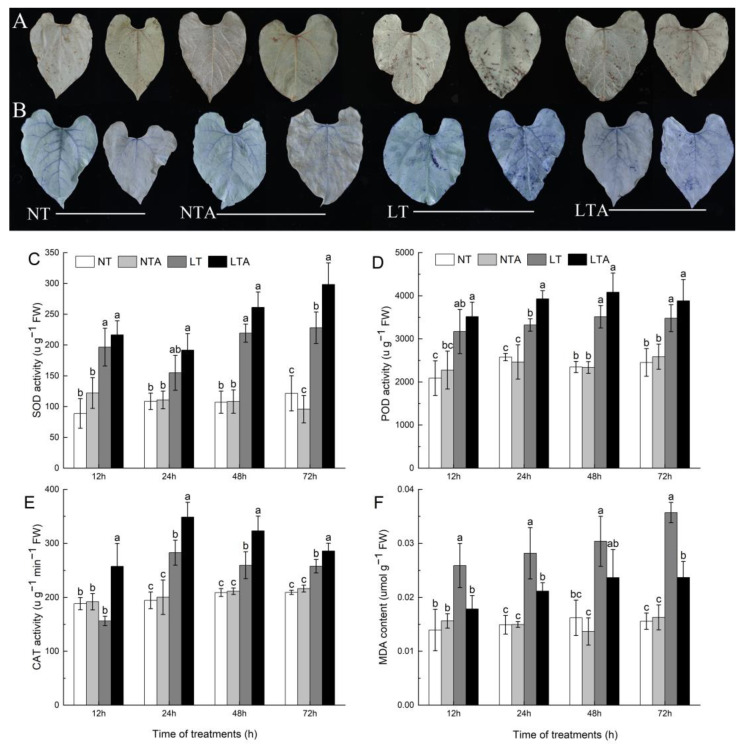
Effects of 5-ALA pretreatment on the accumulation and scavenging of ROS in plants exposed to cold. (**A**) histochemical detection of H_2_O_2_ with 3,3-diaminobenzidine (DAB) staining in leaves 24 h (L) and 72 h (R) after cold treatment; (**B**) Histochemical detection of O^2−^ with nitrotetrazolium blue chloride (NBT) staining in leaves 24 h (L) and 72 h (R) after cold treatment; (**C**) SOD activity; (**D**) POD activity; (**E**) CAT activity; (**F**) MDA content. NT, distilled water plus normal conditions; NTA, 5-ALA plus normal conditions; LT, distilled water plus cold conditions; LTA, 5-ALA plus cold conditions. Each data point represents the mean of three replicate samples, and results are means ± standard deviation. Letters indicate significant differences at *p* < 0.05 by Duncan’s test.

**Figure 4 ijms-24-13189-f004:**
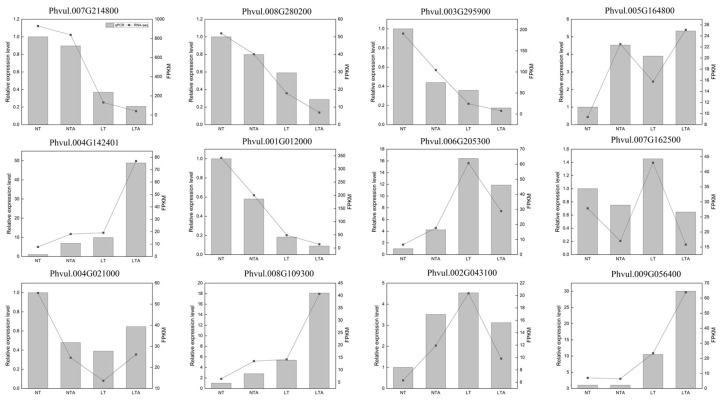
Expression level of randomly selected DEGs. The column diagram and line chart represented the data from qRT-PCR and RNA-seq, respectively.

**Figure 5 ijms-24-13189-f005:**
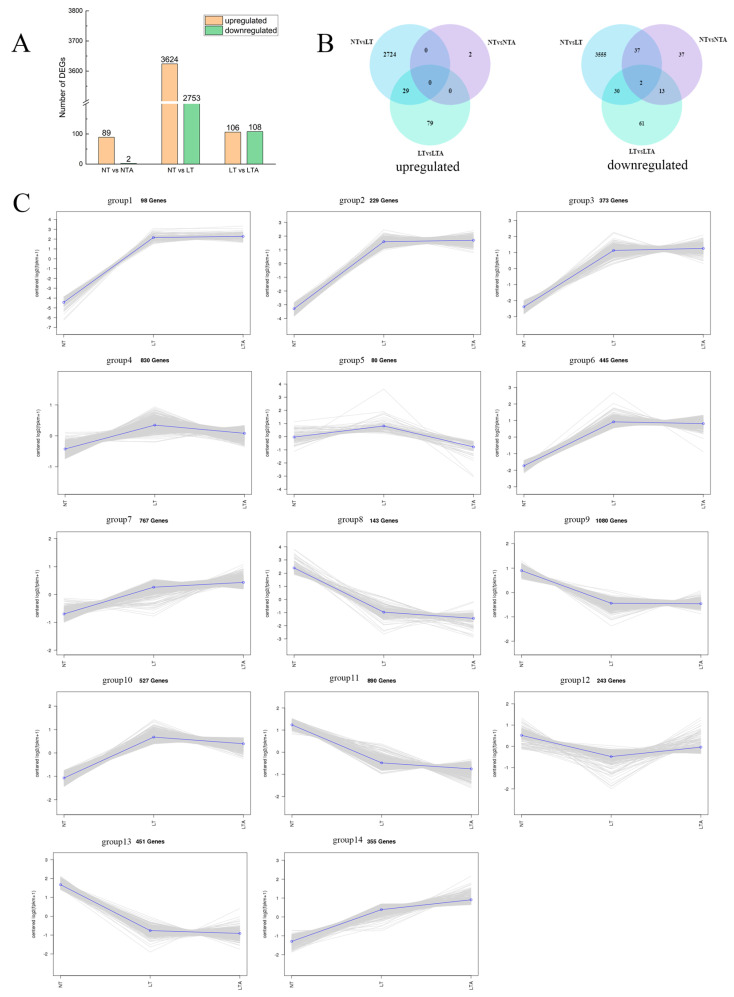
The differentially expressed genes (DEGs) in leaves samples treated with cold or/and 5-ALA pretreatment. (**A**) The number of upregulated and downregulated DEGs; (**B**) Venn diagram of DEGs among the comparison groups; (**C**) Cluster analysis of DEGs by K-means. The gray line represents the expression trend of a gene. The blue line represents the expression trend of these genes.

**Figure 6 ijms-24-13189-f006:**
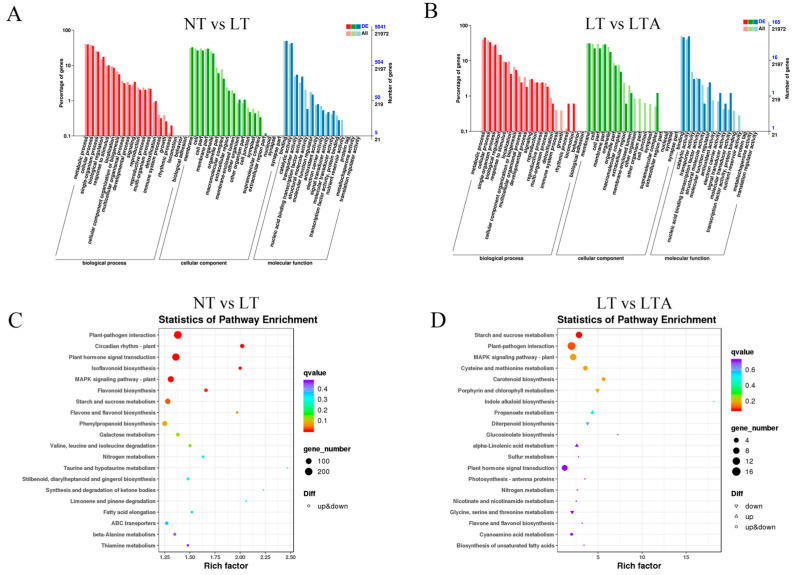
Functional annotations of differentially expressed genes (DEGs) among different treatments. (**A**,**B**) Gene ontology (GO) classification of DEGs; (**C**,**D**) Kyoto Encyclopedia of Genes and Genomes (KEGG) pathway enrichment and bubble chart.

**Figure 7 ijms-24-13189-f007:**
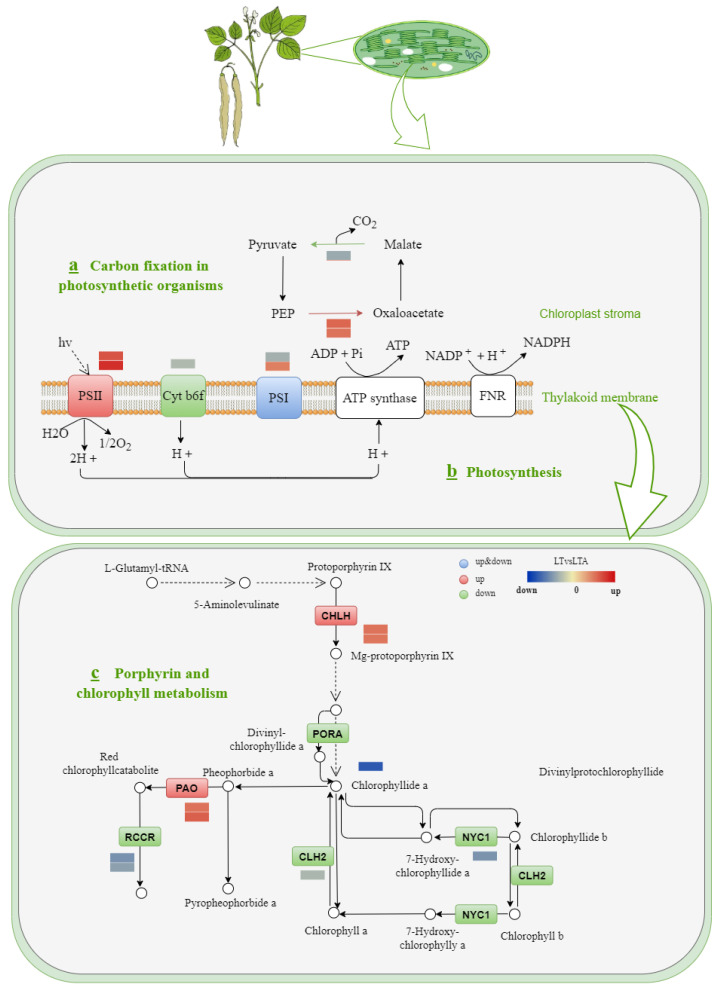
Transcriptional changes of genes involved in plant photosynthesis and porphyrin and chlorophyll metabolism in leaves. Red is related to upregulated genes, green is related to downregulated genes, and blue is related to both upregulated and downregulated genes.

**Figure 8 ijms-24-13189-f008:**
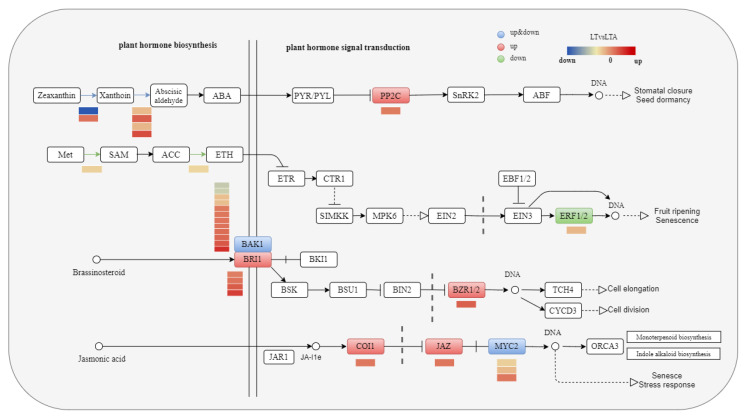
Transcriptional changes of genes involved in plant hormone signal transduction in leaves. Red is related to upregulated genes, green is related to downregulated genes, and blue is related to both upregulated and downregulated genes.

**Figure 9 ijms-24-13189-f009:**
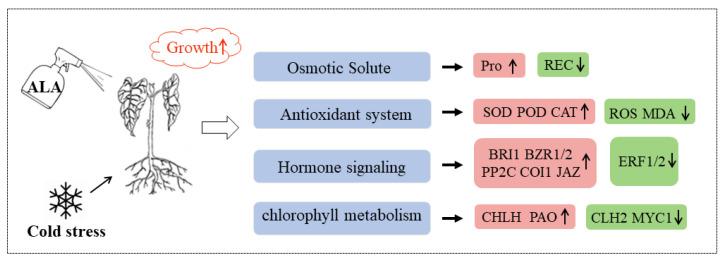
Physiological and molecular regulatory mechanisms by which exogenous 5-ALA application enhances the cold resistance of common bean. Red and green boxes indicate the increases and decreases in the content of substances and gene expression levels, respectively.

## Data Availability

The datasets used and analyzed during the current study are available from the corresponding author on reasonable request.

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
