# Peer review of "5-ALA Improves the Low Temperature Tolerance of Common Bean Seedlings through a Combination of Hormone Transduction Pathways and Chlorophyll Metabolism"

_ijms, 2023, doi:10.3390/ijms241713189_

Round 1

Reviewer 1 Report

The manuscript deals with the assessment of 5-aminolevulinic acid in mitigating cold stress in common bean. The Authors conducted a full spectrum study including physiological, biochemical and genetic analysis. It is very interesting to reconstruct the metabolic pathways as shown in Fig. 7 and 8. The manuscript is a good example of a great effort of scientific work. However, the paper has some flaws. The size of figures should be increased because in current form some inscriptions are unreadable. Numbers referring to the references in the text should be in brackets. Other comments are listed below:

L14-16: add some % changes of examined parameters between treatments

L44-45: indicate that antioxidant enzymes also play a role in plant response to abiotic stress induced by pesticides. This issue is often omitted in scientific studies. For this purpose the Authors may refer to the following references: https://doi.org/10.1007/s00425-022-03838-x

L66-68: explain the abbreviations of the treatments

L137-143: describe some of the results from Fig. 4

L140: in Materials and methods Authors stated that 15 genes were selected for qRT-PCR

L327, 340, 386: species and genus names in italics

L391: how many seeds were sown in one bowl?

L394: 300 μmol·m-2·s-1. Check the superscripts throughout the paper

L426: indicate the concentration of acid ninhydrin

L446: phosphate buffer pH 7.8 is used for SOD activity determination. For CAT and POD buffer pH 7.0 is used.

L453: H2O2

L458: the abbreviations of the treatments should be explained again in Materials and methods

L475: indicate these genes and their functions

L499: do not repeat ‘common bean seedlings’

Author Response

Thank you very much for your attention and suggestions on our manuscript. We tried our best to improve the manuscript, made some modifications to the manuscript, and marked the revised document with revision marks. These changes will not affect the content and framework of the paper. We sincerely thank the reviewers for your enthusiastic work and hope that the corrections will be approved.

-1. L14-16: add some % changes of examined parameters between treatments

-Response: Thank you very much for your suggestion. We have supplemented the % changes of examined parameters between treatments.

-2. L44-45: indicate that antioxidant enzymes also play a role in plant response to abiotic stress induced by pesticides. This issue is often omitted in scientific studies. For this purpose the Authors may refer to the following references: https://doi.org/10.1007/s00425-022-03838-x

-Response: Thank you very much for your suggestion. In the introduction, we described antioxidant enzymes also play a role in plant response to abiotic stress induced by pesticides., and added the reference.

-3. L66-68: explain the abbreviations of the treatments

-Response: Thank you very much for your suggestion. We have supplemented the abbreviations of the treatments in results.

-4. L137-143: describe some of the results from Fig. 4

-Response: Thank you very much for your suggestion. We have added some clarification to the results of Figure 4.

-5. L140: in Materials and methods Authors stated that 15 genes were selected for qRT-PCR

-Response: Please forgive our negligence. We selected 12 genes for qRT-PCR.

-6. L327, 340, 386: species and genus names in italics

-Response: We are very sorry for our incorrect writing. We have revised the species and genus names in italics.

-7. L391: how many seeds were sown in one bowl?

-Response: One seed per nutrient bowl. We have added it to the materials and methods.

-8. L394: 300 μmol·m-2·s-1. Check the superscripts throughout the paper

-Response: Please forgive our negligence. We have checked carefully the superscripts throughout the paper.

-9. L426: indicate the concentration of acid ninhydrin

-Response: Thank you very much for your suggestion. The concentration of acid ninhydrin is 2.5%.

-10. L446: phosphate buffer pH 7.8 is used for SOD activity determination. For CAT and POD buffer pH 7.0 is used.

-Response: Please forgive our negligence. We have added supplemented it here.

-11. L453: H2O2

-Response: Please forgive our negligence. We have checked carefully the subscripts throughout the paper.

-12. L458: the abbreviations of the treatments should be explained again in Materials and methods

-Response: Thank you very much for your suggestion. We have explained the abbreviations of the treatments in experimental design.

-13. L475: indicate these genes and their functions

-Response: Thank you very much for your suggestion. We have supplemented the functions in Table S1.

-14. L499: do not repeat ‘common bean seedlings’

-Response: Thank you very much for your suggestion. We have reduced the use of  ‘common bean seedlings’.

Reviewer 2 Report

In this study, the physiological and transcriptome changes of common bean seedlings in response to cold stress after 5-aminolevulinic acid (5-ALA) pretreatment is explored. Exogenous 5-ALA promote the growth of common bean plants under cold stress, increase the activity of antioxidant enzymes and proline content, decrease the relative conductivity, malondialdehyde, and active oxygen content. 214 differentially expressed genes (DEGs) participate in response to cold stress. Did You find DEGs involved in other types of stress? Did You investigate the changes in ethylene production due to 5-ALA and/or cold treatment? Can 5-ALA be used to decrease stress-effects from other source? Does 5-ALA has contribution with polyamines?

The graphs are sufficiently detailed and easy to expound. The text contains occasional typos, but it is nevertheless easy to read.  The English of the manuscript can to be improved. The literature used is fair enough. Use bracelets for citations in the running text (from 23).

Author Response

Thank you very much for your attention and suggestions on our manuscript. We tried our best to improve the manuscript, made some modifications to the manuscript, and marked the revised document with revision marks. These changes will not affect the content and framework of the paper. We sincerely thank the reviewers for your enthusiastic work and hope that the corrections will be approved.

-1. 214 differentially expressed genes (DEGs) participate in response to cold stress. Did You find DEGs involved in other types of stress?

-Response: We feel extremely sorry that our treatment mainly studied cold stress, so we did not pay special attention to other types of stress in DEGs, which may be paid attention to in future studies.

-2. Did You investigate the changes in ethylene production due to 5-ALA and/or cold treatment?

-Response: We feel extremely sorry that our study did not measure changes in ethylene, but we will continue to explore the relationship between 5-ALA and ethylene.

In other’s research, we know that ethylene plays different roles in response to low-temperature stress in different plants. In tomatoes and cucumbers (Cucumis sativus L.), low-temperature treatment can increase ethylene content in plants. However, in dwarf beans and mung beans (Vigna radiata Linn.), the amount of ethylene synthesis in the body was significantly reduced after low temperature treatment.

In the discussion, we mentioned 5-ALA alleviates the cold stress of common bean by enhancing the accumulation of secondary metabolites and ABA-dependent signal pathways. In addition, ACO, the essential enzyme encoding ETH biosynthetic protein, showed a lower expression level under cold stress, thereby inhibiting ethylene synthesis and the ex-pression of ethylene response factor ERF1B. Consequently, the transcription factor of the ETH signal transduction pathway was downregulated.

In our study, we found a differentially expressed gene encoding ERF1/2 was downregulated in the ethylene signal transduction pathway.

-3. Can 5-ALA be used to decrease stress-effects from other source?

-Response: Priming with 5-ALA can enhance plant tolerance to the subsequent abiotic stresses, including salinity, drought, heat, cold, and UV-B. We have supplemented the corresponding references in the introduction.

-4. Does 5-ALA have contribution with polyamines?

-Response: Due to limited capabilities, we did not find a more systematic answer to this question, but since polyamines and ethylene have a common premise and are competitive, we suspect that 5-ALA will have an impact on multi-pressing. Unfortunately, we will focus on it in subsequent research.

Reviewer 3 Report

A sound, valuable manuscript which brings together both physiological and transcriptional features of cold treated common bean seedlings--specifically, showing the protective effects of 5-ALA against cold treatment.  Some minor concerns noted in the attached manuscript.

I have marked a very small number of English concerns in the attached manuscript.

Author Response

Thank you very much for your attention and suggestions on our manuscript. We tried our best to improve the manuscript, made some modifications to the manuscript, and marked the revised document with revision marks. These changes will not affect the content and framework of the paper. We sincerely thank the reviewers for your enthusiastic work and hope that the corrections will be approved.

-1. I believe that eliminating "the high" would read more clearly.

-Response: Thank you very much for your suggestion. We have eliminated "the high".

-2. Singular subject, so please use corresponding verb forms--that is, "promotes" "increases" and the like.

-Response: Please forgive our negligence. We have corrected it to “promotes”.

-3. Strong abstract--establishes value and importance of your work, and encourages readers to continue to your manuscript!

-Response: Thank you very much for your suggestion. We have supplemented our work’s value and importance in abstract.

-4. Here and throughout your paper--fun to have a "hot link" to each noted reference. However, is that what IJMS expects? Without parentheses around each numerical reference citation, it seems unclear what the numbers might mean.

-Response: Please forgive our negligence. We have bracketed the reference figures.

-5. should your reference 3 be cited here somewhere?

-Response: Please forgive our negligence. We have supplemented the reference 3 and bracketed the reference figures.

-6. Just wondering--did you include a parallel treatment of bean seedlings with 5-ALA in the absence of low-temperature stress? i assume so, based on the results section; however, perhaps you should mention that control here?

-Response: Thank you very much for your suggestion. We have supplemented the normal condition here.

-7. This reviewer assesses that 5-ALA appears to have promoted the growth of bean seedlings under normal temperatures. Perhaps, this should gain mention in your paper? I am aware that this is not the primary premise of your work, but it does seem to me that your NT and NTA treatments were not equivalent in their influences on bean seedlings.

-Response: Thank you very much for your suggestion. We have added the clarification that’’5-ALA promoted the growth of bean seedlings at normal temperature.’’ here.

-8. Perhaps this reviewer is underinformed about the importance of proline content--did you say above why this was an important measurement?

-Response: We have described it in introduction. Proline and other low-molecular-weight solutes act as osmoregulation substances to protect cell osmotic pressure, thereby reducing the damage caused by low-temperature stress in plants.

-9. I am unable to verify this value, based on Figure 2B below.

-Response: Please forgive our negligence. After 72 h of cold stress, the REC of LT plants was 1.69 times that of NT plants.

-10. Article "the" not necessary here.

-Response: We sincerely appreciate your suggestions. We have deleted "the".

-11. Valuable schematic presentation of impacted genes

-Response: Thank you for your appreciation!

-12. Seems to be a very important statement. But how do you know this--did your experiences prove this? Or did prior work by others establish it?

-Response: Please forgive our negligence. We have supplemented the explanation and reference.

-13. Sound series of inferences, based both on your work and that of others.

-Response: Thank you for your appreciation!

-14. Another very helpful schematic presentation!

-Response: Thank you for your appreciation!

-15. Do your figures above indicate that three replicates were used?

-Response: Please forgive our negligence. We used three replicates. We have corrected it to “three”.

-16. Excellent work in bringing together physiological and transcriptional data.

-Response: Thank you again for your appreciation!

-17. Perhaps this is a silly question--just wondering if there is any "real-world" application potential for this work (NOT discounting its value for understanding physiology and "transcriptionality" of beans). In other works, might producers be able to treat their bean seedlings with 5-ALA prior to a cold snap and thus, protect their crop from damage?

-Response: Thank you very much for your suggestion. We have supplemented our conclusion with reflections on this issue. 

Round 2

Reviewer 1 Report

The Authors have improved the manuscript. However, in the revised paper the figures are missing. Besides that I have no more comments.

Author Response

We are sorry that due to our negligence, your review process did not go smoothly. We will add pictures to the revised manuscript. Thank you very much for your patience in review.